# Building Predictive Models for Schizophrenia Diagnosis with Peripheral Inflammatory Biomarkers

**DOI:** 10.3390/biomedicines11071990

**Published:** 2023-07-14

**Authors:** Evgeny A. Kozyrev, Evgeny A. Ermakov, Anastasiia S. Boiko, Irina A. Mednova, Elena G. Kornetova, Nikolay A. Bokhan, Svetlana A. Ivanova

**Affiliations:** 1Budker Institute of Nuclear Physics, Siberian Branch of the Russian Academy of Sciences, 630090 Novosibirsk, Russia; eakozyrev09@gmail.com; 2Institute of Chemical Biology and Fundamental Medicine, Siberian Branch of the Russian Academy of Sciences, 630090 Novosibirsk, Russia; 3Mental Health Research Institute, Tomsk National Research Medical Center of the Russian Academy of Sciences, 634014 Tomsk, Russia; anastasya-iv@yandex.ru (A.S.B.); irinka145@yandex.ru (I.A.M.); ekornetova@outlook.com (E.G.K.); bna909@gmail.com (N.A.B.); 4University Hospital, Siberian State Medical University, 634050 Tomsk, Russia; 5Psychiatry, Addiction Psychiatry and Psychotherapy Department, Siberian State Medical University, 634050 Tomsk, Russia

**Keywords:** schizophrenia, biomarkers, artificial intelligence, machine learning, predictive model, deep neural network, logistic regression, decision trees, support vector machine, k-nearest neighbors

## Abstract

Machine learning and artificial intelligence technologies are known to be a convenient tool for analyzing multi-domain data in precision psychiatry. In the case of schizophrenia, the most commonly used data sources for such purposes are neuroimaging, voice and language patterns, and mobile phone data. Data on peripheral markers can also be useful for building predictive models. Here, we have developed five predictive models for the binary classification of schizophrenia patients and healthy individuals. Data on serum concentrations of cytokines, chemokines, growth factors, and age were among 38 parameters used to build these models. The sample consisted of 217 schizophrenia patients and 90 healthy individuals. The models architecture was involved logistic regression, deep neural networks, decision trees, support vector machine, and k-nearest neighbors algorithms. It was shown that the algorithm based on a deep neural network (consisting of five layers) showed a slightly higher sensitivity (0.87 ± 0.04) and specificity (0.52 ± 0.06) than other algorithms. Combining all variables into a single classifier showed a cumulative effect that exceeded the effectiveness of individual variables, indicating the need to use multiple biomarkers to diagnose schizophrenia. Thus, the data obtained showed the promise of using data on peripheral biomarkers and machine learning methods for diagnosing schizophrenia.

## 1. Introduction

Schizophrenia is a debilitating mental disorder that affects 0.3–0.7% of people worldwide [1]. Currently, the diagnosis of schizophrenia is based solely on the clinical picture of the disease according to the Diagnostic and Statistical Manual of Mental Disorders 5 (DSM-5) [1] or International Classification of Diseases, Tenth Revision (ICD-10) [2]. This leads to difficulties in making a diagnosis, including due to the similarity of the symptoms of schizophrenia with other mental disorders, such as bipolar disorder, depression and autism [3]. Relying solely on clinical symptoms to diagnose schizophrenia can be challenging due to the subjective nature of symptom assessment. In the absence of objective diagnostic tools such as biological or neuroimaging markers, clinical characteristics alone have limited predictive value, as evidenced by lack of response to therapy and disease progression [4,5,6]. The development of artificial intelligence technologies has opened up new possibilities for building predictive models for diagnosing mental disorders. Clinical neuroimaging, language patterns, and mobile phone data are often used to build predictive models of schizophrenia [7,8]. However, clinical data have a high probability of subjectivity. Neuroimaging data is quite difficult to obtain in some cases. Predictive models based on language pattern data require adaptation depending on the country and region and, thus, cannot be universal. Therefore, predictive models based on blood-based biomarkers associated with the pathogenesis of schizophrenia may prove promising.

For many years it was believed that the main biological substrate of schizophrenia is an alteration of the neurotransmission of the neurotransmitters dopamine, glutamate, and serotonin in the CNS [9]. Abnormalities in neural circuits and pathways can disrupt the normal functioning of the brain and contribute to the development of symptoms observed in schizophrenia patients [10,11,12]. However, over the past decade, a large amount of evidence has accumulated that suggests abnormalities of the immune system have an important role in the pathogenesis of schizophrenia, as indicated by peripheral immune alterations [13,14] and neuroinflammation with uncontrolled microglia activation [15,16,17,18,19]. One of the most common molecules that reflect changes in the immune system are cytokines, signaling proteins that regulate the proliferation and activation of immune cells [20]. The primary function of chemokines is to attract immune cells to the inflammation site [21]. Additionally, both chemokines and cytokines may be involved in neurogenesis, neuromodulation, neurotransmission, and can influence behavior [21,22]. In addition to immunoinflammatory mediators, various peripheral growth factors play an important role in the growth, development, differentiation, survival, and migration of cells, including neurons and astrocytes [23]. Cytokines, chemokines, and growth factors may be promising biomarkers of schizophrenia.

While the literature presents a large number of studies of the level of peripheral cytokines associated with schizophrenia, there are slightly fewer data on the peripheral concentration of chemokines. Furthermore, it should be noted that, despite the general trend reflecting an increase in the level of pro-inflammatory cytokines and chemokines associated with this disease, when considering individual indicators, the data are often contradictory [21,24,25,26]. Data on the concentration of growth factors in the peripheral blood in schizophrenic patients, with the exception of a number of neurospecific factors, are limited. Various studies have shown that levels of granulocyte-macrophage colony-stimulating factor (GM-CSF), granulocyte colony-stimulating factor (G-CSF), platelet-derived growth factor with two subunits B (PDGF-BB), and vascular endothelial growth factor A (VEGF-A) were significantly elevated, while the levels of fibroblast growth factor (FGF), epidermal growth factor (EGF), and nerve growth factor (NGF) were significantly reduced in patients with schizophrenia [23,27,28,29].

Given the heterogeneity of the results obtained in previous studies, as well as the multifactorial nature of schizophrenia, we assume that it is not possible to identify one specific disease biomarker using traditional statistical analysis methods. The «biological» diagnosis of schizophrenia should be based on a set of specific markers that may underlie the etiology of the disease. As stated above, to solve this problem, machine learning algorithms can be used to identify subtle patterns in data and build predictive models. In recent years, machine learning methods have been widely used to analyze data from patients with schizophrenia. Approaches to predicting clinical improvement [30], classification of schizophrenia patients and healthy people [31,32], or patients with autism spectrum disorder [33] using machine learning algorithms based on neuroimaging data have been described. A method for diagnosing schizophrenia based on electroencephalography data using a random forest or a deep convolutional neural network has been proposed [34,35,36]. A number of papers based on data from peripheral blood biomarkers using machine learning algorithms have been published. Fernandes et al. (2020) have developed a multi-domain model consisting of peripheral blood immune and inflammatory biomarkers and cognitive biomarkers capable of classifying schizophrenia and healthy individuals with a sensitivity of 84% and a specificity of 81% [37]. An integrated machine learning framework for a discriminative analysis of schizophrenia including gut microbiota data, blood data (inflammation, immunity and oxidative stress), and electroencephalographic data achieved the best results, with an accuracy of 91.7% and an area under curve (AUC) of 96.5% [38]. A machine learning algorithm for diagnosing schizophrenia based on the expression of six genes in peripheral blood showed valuable results with an AUC = 0.993 [39]. Using the machine learning method, complex associations between neuroimmune biomarkers and quality of life in schizophrenia have been shown [40]. An approach to diagnose treatment-resistant schizophrenia based on DNA methylation signature aberration using a machine learning method has also been described [41].

Thus, machine learning and artificial intelligence have become increasingly popular for developing new methods of precision psychiatry [42]. Machine learning algorithms can serve as a valuable tool for clinicians in the prediction, differential diagnosis, and treatment of schizophrenia. However, there are still few studies using peripheral biomarkers to build predictive models for diagnosing schizophrenia. Immune biomarkers including cytokines, chemokines, and growth factors may be particularly useful for constructing predictive models of schizophrenia. Numerous studies have confirmed the association of changes in the concentrations of immune biomarkers with schizophrenia [13,14,26,29]. Data on the concentration of such biomarkers can be obtained using common laboratory methods. Potentially, then, inflammatory biomarkers can provide more objective and reliable indicators for predicting and classifying patients with schizophrenia, and predictive models of schizophrenia based on immune markers could be implemented in clinical practice.

The aim of this work was to develop predictive models based on laboratory data on inflammatory biomarkers for the classification of patients with schizophrenia and healthy individuals. Various clinical scales, such as the Positive and Negative Syndrome Scale (PANSS) [43], have some degree of subjectivity and depend on the experience of the clinician. In contrast, predictive models based on laboratory data on inflammatory biomarkers would have little influence of subjectivity. In the present study, the age of the participants, as a non-subjective variable, was also included in the analysis. Predictive models based on five different architectures were built and tested. To build models, common binary classification algorithms based on logistic regression and deep neural networks were used. Additionally, the efficiency of the algorithms was tested, based on decision trees, support vector machine, and k-nearest neighbors classifier.

## 2. Materials and Methods

### 2.1. Characteristics of Patients and Healthy Individuals Included in the Study

Two hundred seventeen patients with schizophrenia (F20 according to the ICD-10) were examined and recruited from the Mental Health Research Institute of the Tomsk National Research Medical Center of the Russian Academy of Sciences and from the Tomsk Regional Psychiatric Hospital. Inclusion criteria for disease-positive study participants included those with diagnosis of schizophrenia according to ICD-10, age from 18 to 70 years, and absence of signs of acute and chronic infectious-inflammatory and autoimmune diseases. A healthy control group consisted of 90 mentally and somatically healthy individuals, aged from 18 to 70 years, and without signs of acute and chronic infectious-inflammatory and autoimmune diseases. Exclusion criteria for all participants included comorbid neurological and somatic diseases, and dependence on psychoactive substances and alcohol. Any subject could withdraw consent to participate in the study at any time, and the correponding data would be excluded from processing. The severity of schizophrenia symptoms was determined using the Positive and Negative Syndrome Scale (PANSS) [43]. Serum was obtained from peripheral blood samples of all study subjects by centrifugation for 30 min at 2000× *g* at +4 °C. Serum was stored at −80 °C until analysis. Patients were examined in the first days after admission to the hospital.

### 2.2. Multiplex Analysis of the Concentration of Cytokines, Chemokines and Growth Factors in Blood Serum

Determination of cytokine concentration in blood serum was carried out using xMAP technology on analyzers Magpix and Luminex 200 (Luminex, Austin, TX, USA) (Core Facility “Medical Genomics”, Tomsk NMRC). The panel HCYTMAG-60K-PX41 by MILLIPLEX MAP (Merck, Darmstadt, Germany) was used to measure analytes.

The MILLIPLEX MAP system includes antibodies that specifically bind to analytes and are conjugated to xMAP beads. The technology is based on the use of MagPlex-C microspheres—magnetic beads (5.6 to 6.45 μm in diameter) stained with two fluorescent dyes, on the surface of which specific antibodies to the analyzed analytes are localized. Each individual microsphere is identified in the multiplex analyzer and its biological analysis result is quantified based on fluorescent signals.

The detected information is processed by the xPONENT software (Luminex, Austin, TX, USA) with data export to the MILLIPLEX Analyst 5.1 (Merck, Darmstadt, Germany). The concentration results were presented in pg/mL.

### 2.3. Data Preprocessing and Preliminary Statistical Analysis

The data was filtered by removing any rows containing empty cells. To standardize all properties within a [−1, 1] range, a preprocessing step was applied where any value x underwent the transformation 1 − 2/(1 + exp(−(x − mean)/std)), where mean, std are the mean and standard deviation of each variable under analysis in the whole dataset. Preliminary statistical analysis was conducted using the scipy package (1.9.1) in Python language. The Statistica 10 desktop program (StatSoft, Tulsa, OK, USA) was used for preliminary statistical analysis. The significance of differences in the age of patients and healthy individuals was calculated using the Student’s *t*-test. Pearson’s chi-squared test was used to analyze the categorical variable sex.

### 2.4. Building Predictive Models

Predictive models were constructed using open source packages PyTorch (2.0.0), Scikit-learn (1.1.2) and pandas (2.0.1) of Python 3.10.6 programming language. Predictive models were built using algorithms based on logistic regression, deep neural networks, decision trees, support vector machine, and k-nearest neighbors algorithms classifier. Details of building predictive models are presented in Section 3. To check the quality and stability of the constructed models, the cross-validation method was applied using the Scikit-learn package (1.1.2). Sensitivity and specificity parameters for each of the constructed models are presented as the mean and standard deviation calculated for five test samples. The developed code is hosted in the github repository (https://github.com/eakozyrev/schizo2023 (accessed on 18 June 2023)).

## 3. Results

### 3.1. Characterization of the Sample of Patients and Healthy Individuals Used for Classification

The study sample used to build predictive models consisted of 217 patients with schizophrenia and 90 healthy individuals. The study cohorts were comparable in sex and age (Table 1). The male to female ratio was similar in the two groups (52.1/47.9% vs. 50/50%). The mean PANSS total score assessing the severity of clinical symptoms of schizophrenia was 98.7 ± 15.0 points. Other clinical data are presented in Table 1.

The concentration of 37 biomarkers was studied in each participant using multiplex assay. Among the analyzed biomarkers were 22 cytokines: interleukin-(IL)-1α, IL-1β, IL-1 receptor antagonist (IL-1RA), IL-2, IL-3, IL-4, IL-5, IL-6, IL-7, IL-8, IL-9, IL-10, IL-12P40, IL-12P70, IL-13, IL-15, IL-17A, tumor necrosis factor α (TNFα), TNFβ, FMS-like tyrosine kinase 3 ligand (Flt-3L), interferon-(IFN)-α2, and IFNγ. In addition, the concentrations of eight chemokines (MCP-1/CCL2, MIP-1α/CCL3, MIP-1β/CCL4, MCP-3/CCL7, GRO/CXCL1, IP-10/CXCL10, eotaxin/CCL11, MDC/CCL22) were studied. Among the seven growth factors analyzed were the following: epidermal growth factor (EGF), transforming growth factor alpha (TGFα), platelet-derived growth factor-(PDGF)-AA, PDGF-AB/BB, fibroblast growth factor 2 (FGF-2), granulocyte colony-stimulating factor (G-CSF), granulocyte-macrophage colony-stimulating factor (GM-CSF). The obtained data on the concentration of 37 biomarkers were used to build predictive models. Additionally, the variable “age” was included in the analysis. All variables used had minimal risk of subjectivity.

The distributions of the variables used were initially analyzed. Figure 1 shows examples of distributions for two variables that showed signs of some differences. However, variables that differed significantly between patients and healthy individuals were not identified. Therefore, all variables were used to build predictive models.

In some cases, the relationship between variables may be linear. In these cases, linear predictive models can be used. Therefore, linear dependencies between variables were analyzed. A histogram of the linear correlations between the markers used and the main sign that distinguishes healthy and sick subjects is shown in Figure 2. It has been shown that no features have a high correlation value, making it difficult to use linear classification models to differentiate between patients and healthy control groups. The maximal correlation for chemokine MDC/CCL22 is about 0.2. Therefore, more complex models have been applied to classify patients and healthy individuals.

In Figure 3 we show ROC curves for each variable and its opposite, which can be interpreted as reflecting the sensitivity of each variable in classifying the subject. These curves are obtained without any training of any additional parameters, thus, this reflects the degree to which each variable correlates or does not correlate with healthy/patient sign. From Figure 3, it appears that the MDC variable reaches a sensitivity (rate of correct recognition of patients) of approximately 0.79.

As the distribution density of each marker is unknown to us a priori, an empirical sample of the data was used to test the statistical stability of the desired models. Any instability can be due to a small number of events or non-statistical outliers caused by umeasured process factors. Cross-validation tests were performed for each architecture. For cross-validation, the available dataset was divided into five parts to create independent test samples. The five parts were used as follows: the first part was used as the test set, while the other four parts were used as the training set in the following sample sizes: 69 and 176 (healthy individuals and patients, respectively) for training and 21 and 41 (healthy individuals and patients, respectively) for testing. This process was repeated five times, with each part being used as the test set once. This ensured that each sample was used as both training and test data, thus providing a reliable estimate of model performance.

### 3.2. Building a Predictive Model Based on Neural Networks

The following, generally accepted architectures, based on neural networks, were used to build predictive models:Architecture based on logistic regression. It is a linear model that can be represented as f(x) = 1/[1 + exp(−z)], where z = θ_0_ + θ_1_x_1_ + … + θ_n_x_n_, *n* = 38, x are markers-values, and θ are unknown parameters that are determined during model training. As a loss function, which is minimized in the process of training the model, the squared error function MSE is used. For minimization, the gradient descent method was applied with optimization by the Adam method and a penalty function as a regularization. The learning rate hyperparameter was chosen as 0.0003, and the number of epochs is 1500.Architecture based on a deep neural network. The deep neural network used in this study includes four fully connected layers, with 38, 38, 19, 12, and 1 neurons on each layer. The activation function used throughout the network was Relu, except for the last layer, which used the sigmoid function. A total of 2476 parameters were trained. The model was designed to achieve high efficiency with the training samples, as the parameters were selected to maximize the separability of data by class. Although deep neural networks can be prone to overfitting due to a large number of parameters and very small number of samples, we addressed this concern by stopping the parameter optimization process once the mean squared error (MSE) reached 0.05. We also attempted to use dropout regularization to prevent overfitting, but it did not provide significant improvement.

During each epoch of training, we randomly selected the patients, and their sample size was equal to the number of healthy individuals available for training. This ensured that the network was trained using a balanced dataset, with an equal number of events from both classes. The process was repeated for 1500 epochs.

The histograms on the left and right of Figure 4 represent the output distributions for logistic regression and deep neural network, respectively. They provide a summary of the values calculated for all five test samples.

Sensitivity and specificity are two important measures used to evaluate the accuracy of diagnostic tests. High sensitivity is crucial for identifying all patients with a particular condition, while high specificity is crucial for excluding patients who do not have the condition. ROC analysis is used to evaluate the quality of a binary classification. The ROC curves of five models tested on the five independent test sets used in the linear regression (Figure 5A) and deep neural network (Figure 5B) are depicted below. The asterisks indicate the optimal values—the highest value of the ratio of sensitivity and specificity. The profiles of the ROC curves display some scatter due to the limited statistics available. However, the ROC curves for the five independent samples have similar binary classification qualities. Additionally, we generated the ROC curve for the MDC variable. In this scenario, there is no need to train any variables, allowing us to plot the curve for MDC using the entire available dataset. Despite the limited datasets for network training and validation, both linear regression and deep neural network models showcase a slightly higher sensitivity, indicating that the network integrates the cumulative performance of all utilized variables.

Performance metrics were calculated separately for five independent test samples and are presented in Table 2. The deep neural network model has been shown to be slightly more sensitive, but less specific than the logic regression model.

### 3.3. Building a Predictive Model Based on Other Methods of Machine Learning

We also tested other methods, namely, decision tree, support vector classification and the k-nearest neighbors vote, which are all based on different machine learning algorithms and used for classification tasks. A brief description of these architectures is given below.

A decision tree is a tree-like model where each internal node represents a test on an attribute, each branch represents the outcome of the test, and each leaf node represents a class label. Decision trees are easy to interpret. In the present study, instead of the single decision tree, we choose a decision forest with three trees.Support Vector Machine (SVM) is a supervised learning classification algorithm that tries to find a hyperplane that separates two classes with maximum margin. SVC works well for both linearly and nonlinearly separable data. It is suitable for high-dimensional datasets and can handle non-linear decision boundaries. The SVC was used with L2 penalty regularization. For both the decision tree and SVM we weighted events to balance the datasets.k-nearest neighbors classifier (KNN) is a simple and efficient algorithm that is used for multi-class classification. It predicts the class of an unknown data point by looking at the k-nearest data points in the training set. By simple optimization we choose k = 2.

In terms of performance, SVM and KNN are both computationally intensive and can take longer to train on large datasets. Decision trees are relatively faster to train, but they may overfit the data if the tree is too complex. Overall, the choice of algorithm depends on the specific problem requirements and the characteristics of the dataset. However, all three algorithms have been tested.

The performance of these approaches is summarized in Table 3, with standard deviations for each value. Decision trees have been shown to be slightly more sensitive. The KNN algorithm was the least sensitive. However, the specificity was higher for the KNN algorithm (Table 3). Nevertheless, the sensitivity and specificity of these three algorithms were lower than those of the algorithm based on a deep neural network (Table 2).

## 4. Discussion

In this work, five supervised machine learning algorithms were tested to develop predictive models for classifying patients with schizophrenia and healthy individuals. All tested algorithms demonstrated a similar level of sensitivity and specificity. However, the algorithm based on a deep neural network showed slightly higher performance. In the task of classifying patients and healthy people, the algorithm using deep neural networks had a sensitivity of 0.87 ± 0.04 and specificity of 0.52 ± 0.06 (Table 2). The obtained results indicate that neural networks have the ability to generalize input data and exhibit reliable performance.

In addition, we attempted to identify the most impactful internal features to create an optimized model. This involved systematically removing each feature individually or adding features incrementally with subsequent retraining. Through this analysis, we discovered that the MDC/CCL22 marker played a pivotal role in determining the model’s performance. Introducing additional features to the model may not necessarily enhance its performance, as the new input markers would require training new parameters, and if the size of the available data set is insufficient to optimize the expanded model, the resulting performance may be suboptimal. Ultimately, we decided to include all 38 input markers in the model to leverage all available information.

Many attempts have been made to develop predictive and prognostic models for precision psychiatry using machine and deep learning algorithms [7,8]. Neuroimaging, voice and language patterns, mobile phone data, and others are most commonly used for model building [42,44,45]. There are few predictive models based on peripheral biomarker data for diagnosing schizophrenia and other psychiatric disorders [37,39,41]. In one of these models, using data on the expression of six genes in peripheral blood cells, it was possible to achieve specificity = 1.000 and sensitivity = 0.895 [39]; however, the sample (48 schizophrenia patients and 50 healthy subjects) on which this model was built was very small. Other studies have developed models with lower sensitivity and specificity [37,38,41]. Our developed models showed comparable sensitivity (Table 2 and Table 3). However, our models were less specific. This is explained by the relatively small sample size (217 schizophrenia patients and 90 healthy individuals) and rather similar distributions of variables in patients and healthy individuals. It is possible that a larger sample size will allow the development of predictive models with greater specificity.

Taken together, the data obtained showed the promise of using peripheral biomarker data and machine learning methods to build predictive models for classifying patients and healthy individuals. Peripheral biomarkers are readily available for analysis and have become firmly established in routine clinical practice. The processing of biomedical data using artificial intelligence algorithms has the potential to provide new solutions in clinical practice. Predictions obtained using machine learning could become an additional source of information for the diagnosis of schizophrenia and other mental disorders.

### Limitations and Future Directions

This work is preliminary in nature, since the developed predictive models require verification in further studies. The main limitation of this work is related to the small sample size. Due to the limited size of available data (307 events and 38 features), it is not clear whether the higher performance of the neural network compared to other algorithms is associated with statistical limitations or the innate classification potential of features. Additionally, an unbalanced dataset (217 schizophrenia patients and 90 healthy individuals) was used in this work. However, balanced samples were selected from the dataset for training neural networks, thus, this factor did not affect the quality of the built models.

Further research should test the performance of neural networks and other algorithms for diagnosing schizophrenia on large samples. It is also necessary to select the most influential immune biomarkers for classification.

## 5. Conclusions

In this work, five predictive models based on peripheral biomarker concentration data were developed to classify patients with schizophrenia and healthy individuals. Algorithms based on logistic regression, deep neural networks, decision trees, support vector machine, and k-nearest neighbors algorithms showed similar sensitivity and specificity. However, the algorithm based on a deep neural network showed slightly better performance. Using the technique of cross-validation, it was shown that the neural network (consisting of five layers) allows identifying schizophrenia with a sensitivity of 0.87 ± 0.04 and a specificity of 0.52 ± 0.06. Additionally, we have demonstrated that when all variables are combined into a single classifier, the cumulative effect is superior to the performance of individual variables analyzed individually in detecting schizophrenia. Thus, using a single biomarker to diagnose schizophrenia would not be effective. To develop predictive models, it is necessary to use combinations of biomarkers. Data on peripheral biomarkers such as cytokines, chemokines, and growth factors can help build predictive models for diagnosing schizophrenia.

## Figures and Tables

**Figure 1 biomedicines-11-01990-f001:**
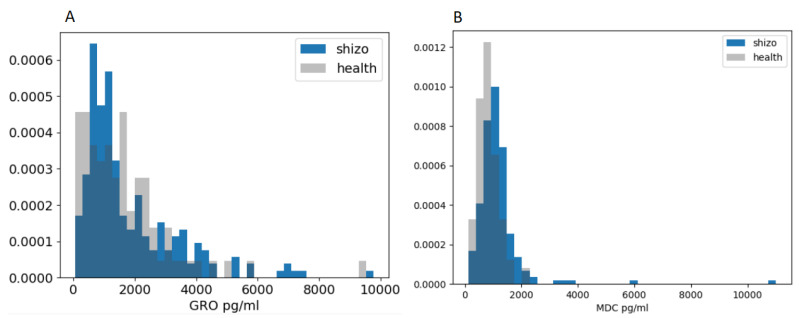
Examples of distributions of GRO (**A**) and MDC (**B**) concentration data in samples of schizophrenic patients and healthy individuals.

**Figure 2 biomedicines-11-01990-f002:**
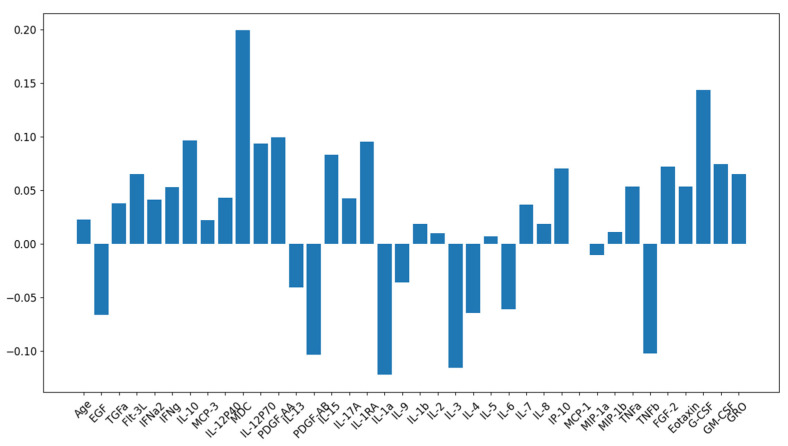
Histogram of linear correlation of markers with healthy/patient sign.

**Figure 3 biomedicines-11-01990-f003:**
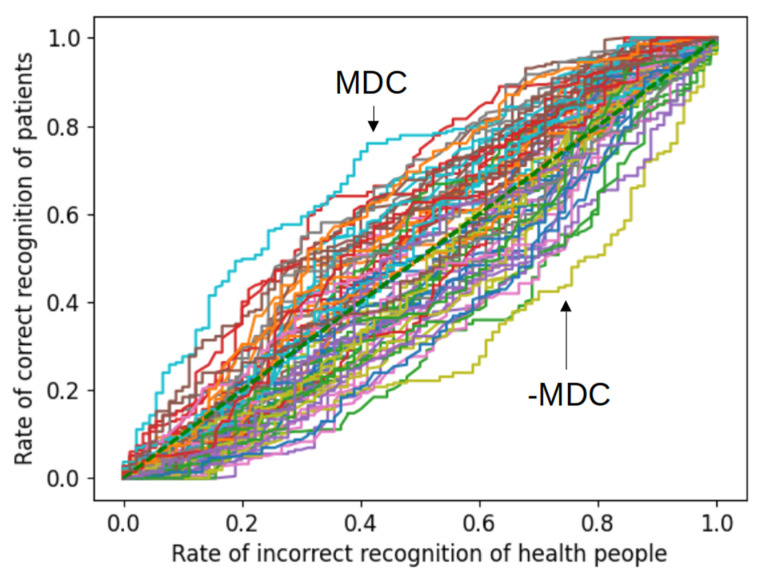
ROC curves for each variable and its opposite, which depict the sensitivity of each variable in classifying the subject matter.

**Figure 4 biomedicines-11-01990-f004:**
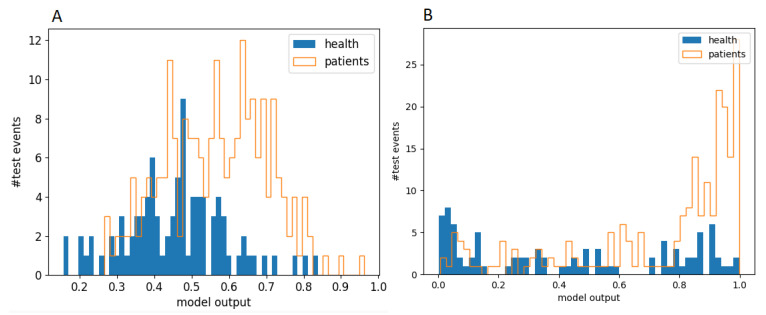
Distributions of predictive model output values based on logistic regression (**A**) and deep neural network (**B**). The histograms summarize the values for all five independent test samples.

**Figure 5 biomedicines-11-01990-f005:**
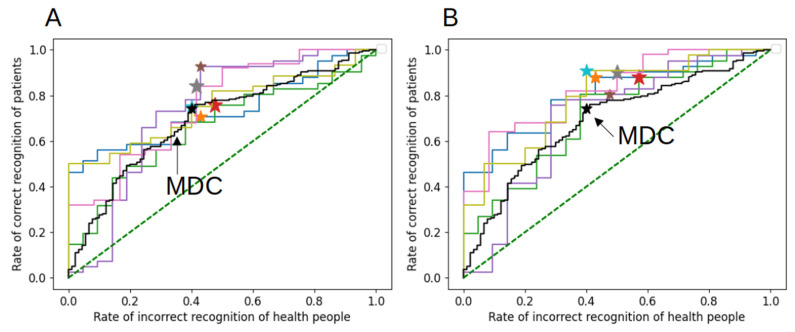
ROC curves of five models on five independent test samples for predictive models based on linear regression (**A**) and deep neural network (**B**).

**Table 1 biomedicines-11-01990-t001:** Demographic and clinical data of the sample of patients and healthy individuals used to build predictive models.

Parameter	Schizophrenia Group	Healthy Group	Significance of Differences
*n*	217	90	N.A.
Sex (F/M), *n*	113/104	45/45	N.S.
Age (Mean ± SD), years	38.3 ± 11.1	37.7 ± 13.2	N.S.
Age at disease manifestation (Mean ± SD), years	25.6 ± 8.1	N.A.	N.A.
Disease duration (Mean ± SD), years	12.8 ± 9.8	N.A.	N.A.
PANSS, positive score (Mean ± SD)	21.0 ± 6.1	N.A.	N.A.
PANSS, negative score (Mean ± SD)	25.5 ± 6.4	N.A.	N.A.
PANSS, general score (Mean ± SD)	52.2 ± 11.7	N.A.	N.A.
PANSS, total score (Mean ± SD)	98.7 ± 15.0	N.A.	N.A.

Note: The significance of the differences was calculated using Student’s *t*-test or Pearson’s chi-squared test (for sex). A *p*-value > 0.05 was considered not significant (N.S.). Abbreviations: N.A.—not applicable, SD—standard deviation.

**Table 2 biomedicines-11-01990-t002:** Comparison of specificity and sensitivity for test samples depending on the predictive model.

Training Samples/Method	Logistic Regression Specificity/Sensitivity	Deep Neural Network Specificity/Sensitivity
Sample No. 1	0.57/0.71	0.57/0.88
Sample No. 2	0.52/0.76	0.43/0.88
Sample No. 3	0.57/0.93	0.52/0.80
Sample No. 4	0.58/0.84	0.50/0.90
Sample No. 5	0.60/0.75	0.60/0.91
Averages	0.57 ± 0.03/ 0.80 ± 0.08	0.52 ± 0.06/ 0.87 ± 0.04

**Table 3 biomedicines-11-01990-t003:** Specificity and sensitivity of predictive models using decision trees, support vector machine, and k-nearest neighbors classifier.

Metric/Method	Decision Tree	SVM	KNN
Specificity	0.465 ± 0.116	0.468 ± 0.153	0.502 ± 0.148
Sensitivity	0.834 ± 0.045	0.803 ± 0.079	0.655 ± 0.045

## Data Availability

The datasets are available on reasonable request to Svetlana A. Ivanova (ivanovaniipz@gmail.com), after approval from the Board of Directors of the Mental Health Research Institute, in accordance with local guidelines and regulations. The code used for analysis is hosted in the github repository (https://github.com/eakozyrev/schizo (accessed on 18 June 2023)).

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
