# Peer review of "Building Predictive Models for Schizophrenia Diagnosis with Peripheral Inflammatory Biomarkers"

_biomedicines, 2023, doi:10.3390/biomedicines11071990_

Round 1

Reviewer 1 Report

1. The standard practice is to utilise neuroimaging, voice and language patterns with relationship to SZ patients. However, there is no justification or any form of new explanation for using data on serum concentrations of cytokines, chemokines and growth factors---some scientific/technical understanding needs to be brought out or justified for.

2. How do the authros deal with the imbalanced dataset 217 schizophrenia patients and 90 healthy individuals? This is very important to detail in the paper.

3. There is no innovation/ new improvements to the AI techniques at all, other than just using standard packages. 

4. Limitations of the study (other than saying using more sample data) should be highlighted. 

-

Author Response

Dear Reviewer,

We thank the reviewer for helpful criticisms and valuable suggestions. We significantly modified our manuscript. We believe that these changes have significantly improved our manuscript.

Below we answer your suggestions point by point. Your comments are in italics. All changes in the manuscript are marked through the Track Changes function of Microsoft Word.

  1. The standard practice is to utilise neuroimaging, voice and language patterns with relationship to SZ patients. However, there is no justification or any form of new explanation for using data on serum concentrations of cytokines, chemokines and growth factors---some scientific/technical understanding needs to be brought out or justified for.

Reply: Indeed, neuroimaging, voice and language pattern data are often used to build predictive models for diagnosing schizophrenia. In addition, clinical data such as PANSS scores are often used. However, clinical data have a high probability of subjectivity and depend on the experience of the clinician. Neuroimaging data is quite difficult to obtain in some cases. Predictive models based on language pattern data require adaptation depending on the country and region and cannot be universal. Therefore, predictive models based on blood-based biomarkers associated with the pathogenesis of schizophrenia may prove promising. Immune biomarkers including cytokines, chemokines and growth factors may be particularly useful for constructing predictive models of schizophrenia. Numerous studies have confirmed the association of changes in the concentrations of immune biomarkers with schizophrenia. In addition, data on the concentration of such biomarkers can be obtained by common laboratory methods. Therefore, predictive models of schizophrenia based on immune markers can be implemented in clinical practice.

We have added this information to the manuscript (please see the Introduction section).

  1. How do the authros deal with the imbalanced dataset 217 schizophrenia patients and 90 healthy individuals? This is very important to detail in the paper.

Reply: Thank you for this really important question. Indeed, imbalanced dataset is used in our work. Please note that N1 refers to the number of patients, while N2 refers to the number of healthy individuals. During each epoch of neural network training, N2 patients were randomly selected from the entire patient pool. This ensured that the network was trained using a balanced dataset, with an equal number of events from both classes. The process is iterated for 1500 epochs. In the BDT and SVM classifiers each patients’ event is weighted by N2/N1 and healthy individuals' event – by N1/N2 factor. Thus, the imbalance of the dataset did not affect the quality of the resulting models.

It is important to note that the accuracy obtained is significantly limited due to the lack of a more comprehensive dataset. Regrettably, there is currently no additional data available. We have made revisions to the article to include this information (please see page 8).

  1. There is no innovation/ new improvements to the AI techniques at all, other than just using standard packages.

Reply: We demonstrated in the article how artificial intelligence techniques can be used to construct a predictive model for diagnosing schizophrenia. We applied commonly used packages and techniques. The novel aspect lies in demonstrating that by combining all variables into a single classifier, a cumulative analysis surpasses the performance of individual variables analyzed separately in identifying schizophrenia. For further clarification, please refer to our modifications in Fig. 3 and Fig. 5, highlighting this assertion.

  1. Limitations of the study (other than saying using more sample data) should be highlighted. 

Reply: We have added a “Limitations and future directions” section to the manuscript where we describe all the limitations of this work.

We hope that based on your suggestions we have improved the manuscript significantly.

Best regards

Authors

Reviewer 2 Report

The present article entitled ‘Classification of schizophrenia patients and healthy individuals based on cytokine, chemokine and growth factor biomarkers data using machine learning algorithms’ by Kozyrev and colleagues delves into the study of neural mechanisms in schizophrenia, with a particular focus on the use of inflammatory biomarkers as predictive indicators. The introduction highlights the importance of understanding neural processes and their contribution to psychiatric disorders like schizophrenia and discusses the complexity of neural interactions, the need for objective measures in diagnosis, and previous research on neural correlates of schizophrenia. Here, the Authors aim to contribute to the existing knowledge base by utilizing advanced analytical techniques to explore the role of neural mechanisms in schizophrenia presenting a research study that investigates the neural mechanisms in schizophrenia, focusing on the use of inflammatory biomarkers as predictive indicators. The study aims to provide valuable insights into the disorder and potentially contribute to advancements in diagnosis and treatment.

In general, I think the idea of this article is really interesting and the authors’ fascinating observations on this timely topic may be of interest to the readers of Biomedicines. However, some comments, as well as some crucial evidence that should be included to support the author’s argumentation, needed to be addressed to improve the quality of the manuscript, its adequacy, and its readability prior to the publication in the present form. My overall judgment is to publish this paper after the authors have carefully considered my suggestions below, in particular reshaping parts of the ‘Introduction’ and ‘Methods’ sections by adding more evidence.

 Please consider the following comments:

I suggest changing the title. In my opinion, in the present form it is wordy/misleading for the readers and not enough informative. An alternative could be “Building Predictive Models for Schizophrenia Diagnosis with Inflammatory Biomarkers”.

Abstract: In my opinion, Authors should consider rephrasing this section. According to the Journal’s guidelines, the Abstract should contain most of the following kinds of information in brief form. Please, consider giving a more synthetic overview of the paper's key points: I would suggest rephrasing the results and conclusion to make them clear for readers to understand.

A graphical abstract that will visually summarize the main findings of the manuscript is highly recommended.

Introduction: The introduction provides a good overview of the topic by highlighting the prevalence of schizophrenia, the limitations of current diagnostic methods, and the role of immune system abnormalities in the pathogenesis of schizophrenia. However, it would be beneficial to further explore the complexity of neural interactions in this disease and describe the intricate network of neural circuits and pathways involved in schizophrenia, by mentioning that abnormalities in these neural circuits can disrupt the normal functioning of the brain and contribute to the development of symptoms observed in schizophrenia patients (https://doi.org/10.3389/fpsyg.2022.1044988; DOI: 10.3390/biomedicines11030945; https://doi.org/10.3389/fpsyt.2023.1225755). Additionally, I would suggest to explain that relying solely on clinical symptoms for schizophrenia diagnosis can be challenging due to the subjective nature of symptom assessment and better discuss how investigating neural markers, such as inflammatory biomarkers, can provide more objective and reliable indicators for predicting and classifying patients with schizophrenia (https://doi.org/10.3390/biomedicines11051465; DOI: 10.3390/biomedicines10122999; https://doi.org/10.3390/biomedicines10092220).

In my opinion, here Authors should clearly state the objective of the study, which is to develop predictive models for classifying patients with schizophrenia using laboratory data on inflammatory biomarkers. I would suggest to emphasize that the focus of the research is to investigate the relationship between neural mechanisms, as indicated by inflammatory biomarkers, and the classification of schizophrenia patients. 

Data preprocessing and preliminary statistical analysis: This section describing data preprocessing and preliminary statistical analysis is quite brief and lacks details. It would be helpful to explain the rationale behind the chosen data preprocessing steps and provide more information on the statistical tests used to analyze the differences between patient and control groups.

In my opinion, I think the ‘Conclusions’ paragraph would benefit from some thoughtful as well as in-depth considerations by the authors, because as it stands, it is very descriptive but not enough theoretical as a discussion should be. Authors should make an effort, trying to explain the theoretical implication as well as the translational application of their research.

In according to the previous comment, I would ask the authors to include a proper and defined ‘Limitations and future directions’ section before the end of the manuscript, in which authors can describe in detail and report all the technical issues brought to the surface.

References: Authors should consider revising the bibliography, as there are several incorrect citations. Indeed, according to the Journal’s guidelines, they should provide the abbreviated journal name in italics, the year of publication in bold, the volume number in italics for all the references. 

I hope that, after these careful revisions, this paper can meet the Journal’s high standards for publication. 

I am available for a new round of revision of this article. 

Best regards,

Reviewer

Minor editing of English language required.

Author Response

Dear Reviewer,

The authors deeply appreciate your thorough analysis and positive evaluation of our manuscript.

Below we answer your suggestions point by point. Your comments are in italics. All changes in the manuscript are marked through the Track Changes function of Microsoft Word.

In general, I think the idea of this article is really interesting and the authors’ fascinating observations on this timely topic may be of interest to the readers of Biomedicines. However, some comments, as well as some crucial evidence that should be included to support the author’s argumentation, needed to be addressed to improve the quality of the manuscript, its adequacy, and its readability prior to the publication in the present form. My overall judgment is to publish this paper after the authors have carefully considered my suggestions below, in particular reshaping parts of the ‘Introduction’ and ‘Methods’ sections by adding more evidence.

  Please consider the following comments:

  • I suggest changing the title. In my opinion, in the present form it is wordy/misleading for the readers and not enough informative. An alternative could be “Building Predictive Models for Schizophrenia Diagnosis with Inflammatory Biomarkers”.

Reply:

Thank you for this suggestion. The title of our article was indeed too long and not informative enough. Therefore, we agree to change the title to the one you suggested with a slight change: “Building Predictive Models for Schizophrenia Diagnosis with Peripheral Inflammatory Biomarkers”.

  • Abstract: In my opinion, Authors should consider rephrasing this section. According to the Journal’s guidelines, the Abstract should contain most of the following kinds of information in brief form. Please, consider giving a more synthetic overview of the paper's key points: I would suggest rephrasing the results and conclusion to make them clear for readers to understand.

Reply: In accordance with your suggestion, we have rephrased the results and the conclusion in Abstract for better clarity for the reader. In particular, we added information that combining all variables into a single classifier showed a cumulative effect that exceeded the effectiveness of individual variables indicating the need to use multiple biomarkers to diagnose schizophrenia. We hope that these changes will make the Abstract more understandable to the reader.

  • A graphical abstract that will visually summarize the main findings of the manuscript is highly recommended.

Reply: We made a graphical abstract and uploaded it to the system.

  • Introduction: The introduction provides a good overview of the topic by highlighting the prevalence of schizophrenia, the limitations of current diagnostic methods, and the role of immune system abnormalities in the pathogenesis of schizophrenia. However, it would be beneficial to further explore the complexity of neural interactions in this disease and describe the intricate network of neural circuits and pathways involved in schizophrenia, by mentioning that abnormalities in these neural circuits can disrupt the normal functioning of the brain and contribute to the development of symptoms observed in schizophrenia patients (https://doi.org/10.3389/fpsyg.2022.1044988; DOI: 10.3390/biomedicines11030945; https://doi.org/10.3389/fpsyt.2023.1225755). Additionally, I would suggest to explain that relying solely on clinical symptoms for schizophrenia diagnosis can be challenging due to the subjective nature of symptom assessment and better discuss how investigating neural markers, such as inflammatory biomarkers, can provide more objective and reliable indicators for predicting and classifying patients with schizophrenia (https://doi.org/10.3390/biomedicines11051465; DOI: 10.3390/biomedicines10122999; https://doi.org/10.3390/biomedicines10092220).

Reply:

Thanks for the recommendation to expand the context of the Introduction section. We agree that schizophrenia is associated with abnormalities in neural circuits and pathways, so we have added the following sentence to the Introduction section: “Abnormalities in neural circuits and pathways can disrupt the normal functioning of the brain and contribute to the development of symptoms observed in schizophrenia patients”.

In addition, we added information that relying solely on clinical symptoms to diagnose schizophrenia can be difficult due to the subjective nature of symptom assessment. We have also added a sentence that “inflammatory biomarkers can provide more objective and reliable indicators for predicting and classifying patients with schizophrenia” (please see the Introduction section).

We have found it possible to cite some of the recommended publications.

  • In my opinion, here Authors should clearly state the objective of the study, which is to develop predictive models for classifying patients with schizophrenia using laboratory data on inflammatory biomarkers. I would suggest to emphasize that the focus of the research is to investigate the relationship between neural mechanisms, as indicated by inflammatory biomarkers, and the classification of schizophrenia patients.

Reply: Thank you for this suggestion. We have adjusted the aim of the study to the following: “The aim of this work was to develop predictive models based on laboratory data on inflammatory biomarkers for the classification of patients with schizophrenia and healthy individuals”.

  • Data preprocessing and preliminary statistical analysis: This section describing data preprocessing and preliminary statistical analysis is quite brief and lacks details. It would be helpful to explain the rationale behind the chosen data preprocessing steps and provide more information on the statistical tests used to analyze the differences between patient and control groups.

Reply: Great suggestion. We used the following transformation for preprocessing: 1 - 2/(1 + exp(-(x - mean) / std)). This is a commonly used and convenient method of data transformation. The key part of preprocessing is the normalization that is required to bring various data in various units of measurement and ranges of values to a single form that will allow them to be compared with each other and make its combinations. We have included an explanation in the manuscript (please see the part 2.3. Data preprocessing and preliminary statistical analysis).

  • In my opinion, I think the ‘Conclusions’ paragraph would benefit from some thoughtful as well as in-depth considerations by the authors, because as it stands, it is very descriptive but not enough theoretical as a discussion should be. Authors should make an effort, trying to explain the theoretical implication as well as the translational application of their research.

Reply: We agree that the Conclusion section was too descriptive. The main theoretical and translational result of our work is that when all variables are combined into a single classifier, the cumulative effect is superior to the performance of individual variables analyzed individually in detecting schizophrenia. Thus, using a single biomarker to diagnose schizophrenia would not be effective. To develop predictive models, it is necessary to use combinations of biomarkers. Data on peripheral biomarkers such as cytokines, chemokines, and growth factors can help build predictive models for diagnosing schizophrenia. We have added these considerations to the Conclusion section.

  • In according to the previous comment, I would ask the authors to include a proper and defined ‘Limitations and future directions’ section before the end of the manuscript, in which authors can describe in detail and report all the technical issues brought to the surface.

Reply: We have added a “Limitations and future directions” section to the manuscript where we describe all the limitations of this work and areas for further research.

  • References: Authors should consider revising the bibliography, as there are several incorrect citations. Indeed, according to the Journal’s guidelines, they should provide the abbreviated journal name in italics, the year of publication in bold, the volume number in italics for all the references.

Reply: Thank you for noticing this. To compile the bibliographic list, we used Zotero and a special library for citing according to MDPI requirements. The bibliography was generated automatically. Perhaps there were errors in the meta-data of the articles. Therefore, there were inaccuracies in the bibliography. We have adjusted the bibliography.

We hope that based on your suggestions we have improved the manuscript significantly.

Best regards

Authors

Round 2

Reviewer 1 Report

The authors have addressed most of my concerns. After the changing of certain wording, the revised manuscript presents a more realistic nature of the results obtained. I suggest for the authors to include a statement in the Limitations that the study is still of preliminary nature and require a larger sample size to further validate the results. The keyword here is "preliminary study" so as to be fair to the readers. 

-

Author Response

Dear Reviewer,

Thank you very much for your efforts in reviewing our manuscript.

We agree that our study is preliminary in nature. In future studies, larger samples should be used to validate the results. Therefore, we have added the following sentence to the Limitations and future directions section to emphasize the preliminary nature of our work: “This work is preliminary in nature, since the developed predictive models require verification in further studies”.

Thanks for your thoughtful suggestions.

Best regards

Authors

Reviewer 2 Report

The authors did an excellent job clarifying all the questions I have raised in my previous round of review. Currently, this paper is a well-written, timely piece of research and provides a useful description of the neural mechanisms underlying schizophrenia and the potential use of inflammatory biomarkers as predictive indicators.

Overall, this is a timely and needed work. It is well-researched and nicely written, with a good balance between descriptive and narrative text.

I believe that this paper does not need a further revision, therefore the manuscript meets the Journal’s high standards for publication.

I am always available for other reviews of such interesting and important articles.

Reviewer

Author Response

Dear Reviewer,

Thank you for appreciating our manuscript.

We appreciate your time spent with our manuscript.

Thank you for your valuable and thoughtful suggestions.

Best regards

Authors